# An unexpected role for the yeast nucleotide exchange factor Sil1 as a reductant acting on the molecular chaperone BiP

Kevin D Siegenthaler, Kristeen A Pareja, Jie Wang, Carolyn S Sevier*

Department of Molecular Medicine, Cornell University, Ithaca, United States

**Abstract** Unfavorable redox conditions in the endoplasmic reticulum (ER) can decrease the capacity for protein secretion, altering vital cell functions. While systems to manage reductive stress are well-established, how cells cope with an overly oxidizing ER remains largely undefined. In previous work (Wang et al., 2014), we demonstrated that the chaperone BiP is a sensor of overly oxidizing ER conditions. We showed that modification of a conserved BiP cysteine during stress beneficially alters BiP chaperone activity to cope with suboptimal folding conditions. How this cysteine is reduced to reestablish 'normal' BiP activity post-oxidative stress has remained unknown. Here we demonstrate that BiP's nucleotide exchange factor – Sil1 – can reverse BiP cysteine oxidation. This previously unexpected reductant capacity for yeast Sil1 has potential implications for the human ataxia Marinesco-Sjögren syndrome, where it is interesting to speculate that a disruption in ER redox-signaling (due to genetic defects in *SIL1*) may influence disease pathology.

*For correspondence: css224@cornell.edu

**Competing interests:** The authors declare that no competing interests exist.

## Introduction

In eukaryotes, the oxidizing environment of the endoplasmic reticulum (ER) facilitates the folding and secretion of approximately a third of the cellular proteome. Protein flux through the ER varies widely, and transient increases in oxidative folding both deplete reduced glutathione and generate hydrogen peroxide, which can disrupt protein structure, folding, and secretion (*Bulleid and Ellgaard, 2011*; *Kakihana et al., 2012*). We have shown in yeast that a conserved cysteine in the nucleotide-binding domain of the Hsp70 chaperone BiP (Kar2) senses alterations in levels of both glutathione and peroxide in the ER. As ER levels of these small molecules rise, the BiP cysteine becomes oxidized, converting the normally ATP-driven chaperone into an ATP-independent protein holdase (*Wang et al., 2014*; *Wang and Sevier, 2016*). A similar system has been proposed for mammals, where formation of an intramolecular disulfide bond in the presence of increased oxidants augments BiP chaperone function (*Wei et al., 2012*). The increased chaperone activity of oxidized BiP is proposed to promote cell survival by limiting polypeptide aggregation during suboptimal folding conditions (*Wang et al., 2014*).

A hallmark of thiol-redox switches is their reversibility, which allows for a reversion to 'normal' activity when oxidative stress subsides. Yet how BiP reduction is achieved within cells has remained unclear. The ER contains multiple members of the thioredoxin superfamily with the capacity to reduce oxidized thiols; yet the relatively buried location of the redox-sensitive cysteine in BiP suggests that BiP is a poor candidate substrate for these reductases. Here we identify Sil1, BiP's nucleotide exchange factor (NEF), as an unexpected reductant of oxidized BiP. We propose that a redox-active cysteine pair within a flexible N-terminal polypeptide domain of Sil1 facilitates reduction of the relatively buried BiP cysteine.

## Results and discussion

A role for Sil1 in controlling the redox state of the BiP cysteine emerged from a genetic screen designed to isolate yeast BiP alleles that increase the viability of cells exposed to oxidative ER stress conditions. This screen took advantage of our prior observations that (i) a yeast strain unable to undergo BiP oxidation (a *kar2-C63A* strain) was inviable when cells were subject to oxidative ER stress and (ii) an ectopic BiP allele that functionally mimics oxidized BiP allows for robust growth of the compromised *kar2-C63A* strain (*Wang et al., 2014*). Building upon these phenotypes, we randomly mutagenized BiP and screened for alleles that allowed for robust growth of a *kar2-C63A* yeast strain expressing a hyper-active mutant of the oxidoreductase Ero1 (Ero1*), which we used as proxy for physiological ER oxidative stress (*Sevier et al., 2007*). We aimed to isolate BiP mutants that either stabilized the oxidized BiP form or phenotypically mimicked the oxidized form, without necessarily impacting BiP oxidation. Our screen identified a BiP-K314E mutant allele. BiP is an essential gene in yeast. We observed that the K314E mutation does not compromise essential BiP activity; a BiP-K314E allele can support cell viability as the sole cellular BiP (*Figure 1A*). Yet, in keeping with the original screen design, a strain containing a BiP-K314E mutant was able to more efficiently manage oxidative stress, exhibiting a greater resistance to the small molecule oxidant diamide (*Figure 1A*).

The redox-active BiP cysteine (Cys63) and Lys314 are both located in the BiP ATPase domain; Cys63 is relatively hidden within a cleft that forms the nucleotide-binding pocket while Lys314 is surface exposed, found at the interface formed between BiP and its NEF Sil1 (*Figure 1—figure supplement 1*) (*Yan et al., 2011*). Although mutation of the surface exposed Lys314 could alter the local cysteine environment to modulate cysteine oxidation, we were more intrigued by the possibility that introduction of a negative charge a position 314 may weaken the interaction between BiP and Sil1, which could beneficially alter BiP activity during oxidative stress. Using a GST-pulldown assay, we confirmed a clear disruption in the physical association between Sil1 and recombinant BiP-K314E (*Figure 1—figure supplement 2A*). A disruption in the interaction between BiP-K314E and Sil1 in vivo was also implied by the equivalent phenotypes observed with BiP-K314E and *sil1Δ* alleles. BiP utilizes two NEFs, and a strain lacking both NEFs (*lhs1Δ sil1Δ*) is inviable (*Tyson and Stirling, 2000*); a BiP-K314E allele behaved like a *sil1Δ* allele, showing inviability in combination with *lhs1Δ* (*Figure 1—figure supplement 2B*). Given these data, we initially speculated that the resistance to oxidant observed for the BiP-K314E alleles was due to the loss of NEF interaction, letting BiP dwell longer in an ADP/peptide-bound state, enhancing holdase activity like oxidized BiP (*Wang et al., 2014*). Yet, while such a mechanism may contribute to some of the beneficial impact of the BiP-K314E allele during stress, we observed that the resistance to diamide conferred by the K314E mutation was largely abolished upon mutation of BiP Cys63 (a C63A-K314E mutation; *Figure 1A*). Curiously, these data implied that the K314E alteration may influence BiP cysteine oxidation despite the lack of proximity between Lys314 and Cys63.

To directly test the influence of the K314E mutation on BiP cysteine oxidation, we performed a biotin-switch procedure that allows for the conversion of oxidized cysteine adducts to biotinylated cysteines, which are readily detectable with an avidin probe (*Figure 1B*). Yeast BiP contains a single cysteine (Cys63), and any avidin signal in this switch assay has been traced to Cys63 oxidation (*Wang et al., 2014*). Confirming our prior results, approximately two-fold more oxidized BiP was recovered from cells grown under conditions of ER stress (*Figure 1C*) (*Wang et al., 2014*). Strikingly, a BiP-K314E allele further enhanced the recovery of oxidized BiP from stressed cells (*Figure 1C*). If the enhanced oxidation of BiP-K314E is a consequence of disrupted Sil1 binding, a similar increase in BiP oxidation levels should be observed in cells lacking Sil1. Indeed, a *sil1Δ* strain not only accumulated more oxidized BiP than a wild-type strain under stress but also showed a higher basal level of oxidized BiP in the absence of stressor (*Figure 1D*). A *sil1Δ* strain also exhibited an increased ability to survive in the presence of diamide, relative to a wild-type strain (*Figure 1E*); the increased resistance to diamide for a *sil1Δ* strain mirrors the increased diamide resistance observed for a strain containing BiP-K314E (*Figure 1A*). Together, these data are consistent with a model wherein a disruption in the association between BiP and Sil1 in cells results in an increased level of oxidized BiP under stress conditions. We propose that it is the accumulation of more oxidized BiP in these cells that contributes towards the increase in cell survival observed for these strains when grown in the presence of oxidant (diamide). In keeping with this model, the resistance of a *sil1Δ* strain to diamide

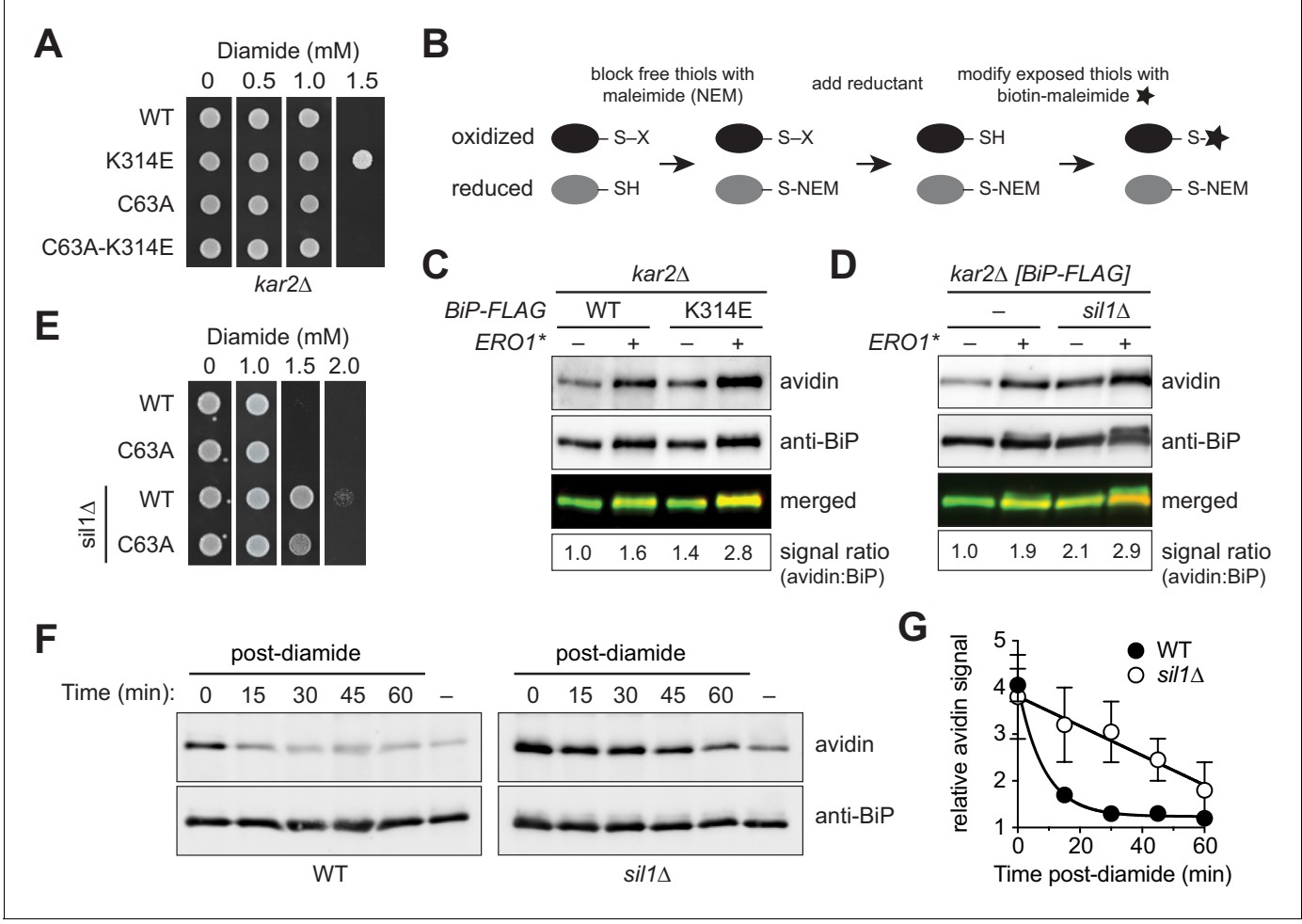

**Figure 1.** Sil1 regulates BiP oxidation state in cells. (**A**) Yeast strains (CSY289, 290, 612, 689) were spotted onto SMM plates containing 0–1.5 mM diamide and incubated for 2 d at 30℃. (**B**) Schematic for the biotin-switch procedure. (**C**) Yeast strains deleted for endogenous BiP (*kar2Δ*) containing plasmids encoding FLAG-tagged BiP were assayed for oxidized BiP levels using the biotin-switch protocol. Oxidative stress was generated by overexpression of Ero1*. BiP was immunoprecipitated, and total and oxidized BiP were detected by Western blotting. The relative levels of oxidized BiP are expressed as the ratio of the intensity of the avidin and anti-BiP signals. The signal ratio was set to 1.0 for wild-type cells grown in the absence of Ero1*. (**D**) Lysates were prepared from the indicated yeast after Ero1* induction. Oxidized BiP levels were detected and quantified as in C. (**E**) Yeast strains (CSY5, 275, 448, 449) were spotted onto YPD plates containing 0–2.0 mM diamide and were incubated for 2 d at 30℃. (**F**) Cells were treated with 5 mM diamide for 15 min, diamide was removed, and cells were returned to 30℃ until harvest. Oxidized BiP levels were determined as in C. (**G**) Plot of the averaged quantified data ± SEM from F and a second independent experiment using the same protocol. For each strain, the signal ratio was set to 1.0 for cells grown without diamide.

The following figure supplements are available for figure 1:

**Figure supplement 1.** BiP-Sil1 structure.

**Figure supplement 2.** A BiP K314E mutation disrupts Sil1 binding.

was lessened when the BiP cysteine was mutated to alanine (*Figure 1E*). However, it is important to note that the increased diamide resistance observed for the *sil1Δ* strain cannot be attributed exclusively to a role for Sil1 in modulating BiP's redox state; a *sil1Δ kar2-C63A* strain displays more resistance to diamide than a *kar2-C63A* strain (*Figure 1E*). These data suggest that there is also some benefit for loss of *SIL1* during oxidative stress independent of BiP cysteine oxidation; we suggest that changes in BiP function in cells as a consequence of a loss of Sil1 NEF activity also facilitate diamide resistance.

The increased level of oxidized BiP observed in both a BiP-K314E mutant and *sil1Δ* strain implies that the normal association of BiP and Sil1 either (i) inhibits adduct formation or (ii) facilitates adduct reduction. We have shown previously that when stress conditions subside, the BiP cysteine-adduct is reduced (removed) with a half-life of less than 10 min (*Wang and Sevier, 2016*). To determine if Sil1 facilitates BiP reduction, we monitored whether an absence of Sil1 slowed the removal of the BiP cysteine-adduct post-oxidative stress. Utilizing the biotin-switch assay, we observed a rapid decrease in oxidized BiP levels in a wild-type strain post-oxidant removal ($t_{1/2}$ ~6 min) (*Figure 1F,G*). In contrast, the stability of the BiP cysteine-adduct was markedly enhanced in a strain lacking Sil1 (*sil1Δ*), demonstrating a half-life of greater than 45 min (*Figure 1F,G*). Sil1 has been implicated in the retro-translocation of cholera toxin (*Williams et al., 2015*). However, the relatively constant and similar levels of total BiP in the wild-type and *sil1Δ* strains suggests that changes in ER-associated degradation do not account for the stabilization of oxidized BiP. Of note, the level of oxidized BiP in a *sil1Δ* strain was restored to wild-type levels 1 hr post-oxidant removal (*Figure 1F,G*), demonstrating that other (slower) mechanisms exist for BiP reduction in the absence of Sil1. The presence of compensatory reduction systems is consistent with the viability of the *sil1Δ* strain. If Sil1 were the sole facilitator of BiP reduction, loss of Sil1 activity might be expected to be lethal to cells; we have shown previously that a strain containing a mimetic allele of constitutively oxidized BiP (as the sole copy of cellular BiP) is inviable under non-stress conditions (*Wang et al., 2014*).

We initially reasoned that the documented ability of Sil1 to ratchet open the nucleotide-binding cleft (*Yan et al., 2011*) could allow a reductant access to the cysteine. However, we were struck by the presence of a pair of cysteines in the N-terminal domain of Sil1 separated by four intervening residues (Cys-52 and Cys-57), which suggested the intriguing alternative that Sil1 itself could be an oxidoreductase. These cysteine residues are absent in the BiP-Sil1 structure (*Yan et al., 2011*), yet it is easy to envision how a polypeptide sequence extending from the most N-terminal residue of Sil1 in the structure could reach into the BiP nucleotide-binding cleft to allow for thiol-disulfide exchange between Sil1 and oxidized BiP (*Figure 1—figure supplement 1*).

To determine if Sil1 has the capacity to reduce oxidized BiP, we purified recombinant BiP and Sil1 from bacteria and assayed for Sil1 activity as a reductant in vitro. In cells, BiP is oxidized by peroxide and glutathione, forming either a sulfenic acid or glutathione adduct (*Wang et al., 2014*; *Wang and Sevier, 2016*). To facilitate monitoring of oxidized BiP in vitro, we reacted BiP with Ellman's reagent (DNTB), which results in a BiP-TNB disulfide linkage similar to the BiP-glutathione disulfide link; a protein-TNB substrate has also been utilized as an effective substrate for an enzyme that reverses sulfenic acid adducts (*Depuydt et al., 2009*). A BiP-TNB adduct is colorless, yet reduction of BiP liberates the TNB anion, allowing for spectroscopic monitoring (*Figure 2A*). Strikingly, Sil1 reduced the otherwise stable BiP-TNB adduct (*Figure 2B*). Sil1 reducing activity required the presence of either Cys52 or Cys57; a C52A-C57A mutant showed no capacity to facilitate BiP reduction (*Figure 2B*). Interestingly, the presence of a single N-terminal cysteine (either Cys-52 or Cys-57) was sufficient to release TNB, demonstrating that each cysteine can act as the attacking nucleophile (*Figure 2B*). Such a mechanism is distinct from that used by the well-characterized reductase thioredoxin, where only the first cysteine in the active site Cys-X-X-Cys motif is able to serve as an attacking nucleophile (*Lu and Holmgren, 2014*). However enzymes containing redox-active cysteine pairs that do not adopt a thioredoxin fold have shown a behavior similar to what is seen with Sil1, including the ER-localized oxidase Erv2 (*Gross et al., 2002*). Sil1 contains two additional cysteines within the armadillo repeats (Cys-203 and Cys-373); we observed that these cysteines were dispensable for Sil1 reducing activity (*Figure 2B*).

Disruption of residues at the contact sites formed between BiP and Sil1 also hindered BiP-TNB reduction. Sil1 His163 has been shown to be critically involved in the association of Sil1 with BiP (*Yan et al., 2011*), and we observed that a Sil1-H163E mutant was unable to effectively reduce BiP-TNB (*Figure 2B*). Similarly, mutation of BiP Lys314 (a BiP-K314E mutant) lessened the removal of the TNB adduct by wild-type Sil1 (*Figure 2—figure supplement 1*). Some modest release of TNB from oxidized BiP-K314E by Sil1 was observed (*Figure 2—figure supplement 1*), and we suggest that the limited reductant activity observed with BiP-K314E reflects some productive association of BiP-K314E and Sil1 at the high BiP:Sil1 ratio required for this single-turnover assay. Notably, the release of TNB from BiP-K314E by Sil1 was lost when the Sil1 N-terminal cysteines were mutated to alanine (*Figure 2—figure supplement 1*).

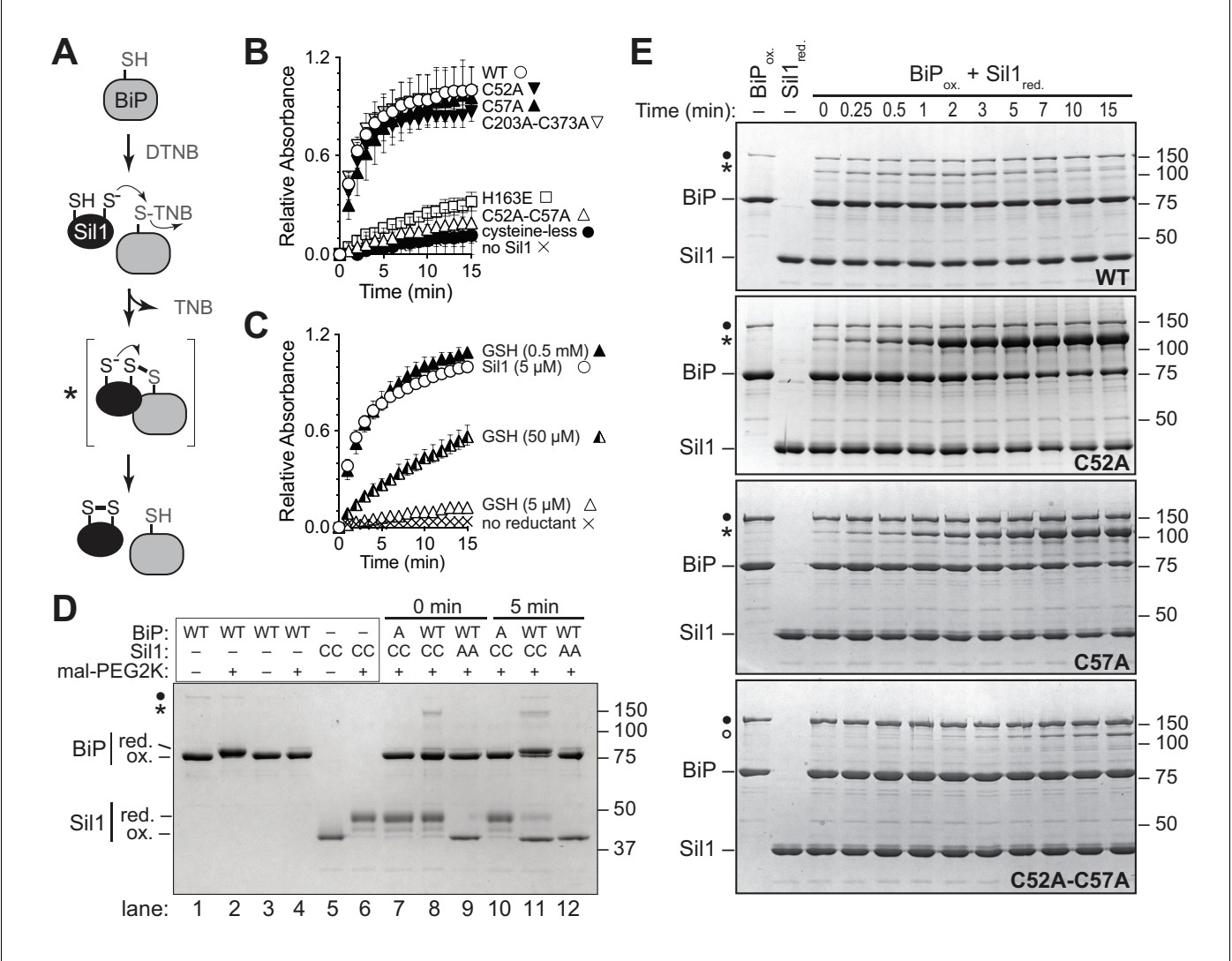

**Figure 2.** Sil1 N-terminal cysteines facilitate reduction of oxidized recombinant BiP in vitro. (A) Schematic for monitoring Sil1 activity as a BiP-cysteine reductant. (B) Reduction of recombinant BiP oxidized by DTNB (BiP-TNB), in the presence of recombinant wild-type or mutant Sil1, was monitored by following the change in absorbance at 412 nm (indicative of TNB release) over time. (C) BiP-TNB reduction by reduced glutathione (GSH) or Sil1 was monitored as in B. Data in B and C represent the mean values from three independent experiments; error bars depict the range. (D) Thiol-disulfide exchange between recombinant wild-type BiP (WT) or a cysteine-less BiP-C63A mutant (A) and Sil1-C203A-C373A (with the N-terminal cysteines; CC) or a cysteine-less Sil1 (lacking the N-terminal cysteines; AA) was monitored by following the presence or absence of free thiols in both proteins. BiP and/or Sil1 were incubated for the indicated times, and reactions were quenched with the addition of the thiol-modifying agent mal-PEG2K. Samples were separated by non-reducing SDS-PAGE and visualized with Coomassie blue. Lanes 1–6 are shown as mobility controls. Lanes 1 and 2 contain BiP that was not reacted with DTNB; all other lanes include BiP incubated with DNTB prior to the addition of Sil1, and later mal-PEG2K. Labels indicate disulfide-linked BiP-Sil1 (asterisk) and BiP-BiP (filled circle) species. (E) Coomassie-stained gels follow recombinant BiP-TNB reaction with wild-type and mutant Sil1 proteins. Samples were quenched at the indicated times with N-ethylmaleimide (NEM) and separated by non-reducing SDS-PAGE. A BiP-Sil1 mixed-disulfide species (asterisk) accumulates with Sil1-C52A and Sil1-C57A. A second BiP-Sil1 species maintained in the absence of the N-terminal Sil1 cysteine pair is noted with an open circle.

The following figure supplements are available for figure 2:

**Figure supplement 1.** Oxidized BiP-K314E is a relatively poor substrate for Sil1.

**Figure supplement 2.** High molecular weight protein species are resolved by reducing SDS-PAGE.

**Figure supplement 3.** BiP's cysteine is required to form the disulfide-bonded species observed with Sil1-C57A.

*Figure 2 continued on next page*

*Figure 2 continued*

**Figure supplement 4.** High molecular weight species observed under non-reducing conditions are absent when BiP is incubated with a cysteine-less Sil1 mutant.

**Figure supplement 5.** High molecular weight species observed under non-reducing conditions (and enhanced when Sil1 contains a single N-terminal cysteine) contain both Sil1 and BiP.

To test Sil1 activity relative to a characterized reductant, we compared the activity of Sil1 to reduced glutathione (GSH). Surprisingly, GSH showed negligible activity when tested at an equivalent concentration to Sil1 (*Figure 2C*). In fact, a 100-fold excess of GSH was required to recapitulate comparable Sil1 activity (*Figure 2C*). We expect that the high affinity reported for Sil1 towards BiP likely accounts for the increased reducing capacity observed for Sil1 relative to GSH (*Yan et al., 2011*). Given the abundant (millimolar) amounts of glutathione present in the ER, it remains an open question whether GSH contributes to BiP reduction in vivo.

We expect that BiP-TNB reduction by Sil1 proceeds through a dithiol-disulfide exchange reaction, wherein recovery of the BiP cysteine thiol is coincident with oxidation of the N-terminal Sil1 cysteines (*Figure 2A*). To confirm that oxidation of the Sil1 cysteines is concomitant with BiP reduction, we monitored the Sil1 cysteine redox state using a 2-kD maleimide-PEG reagent (mal-PEG2K), which will react with reduced thiols resulting in a mobility shift detectable by SDS-PAGE. In order to specifically follow the N-terminal cysteine pair, we used a Sil1-C203A-C373A mutant. We first determined the relative mobility on a SDS-polyacrylamide gel for the oxidized and reduced forms of BiP and Sil1 after mal-PEG2K addition. Modification of the free thiol in reduced BiP with mal-PEG2K resulted in a slower migrating species (*Figure 2D*, lane 2) that could be clearly distinguished from oxidized BiP (BiP-TNB), which was not susceptible to alkylation by mal-PEG2K and migrated similarly to BiP not treated with mal-PEG2K (*Figure 2D*, lane 4 versus 3). Likewise, alkylation of the two free thiols in reduced Sil1 resulted in a slow migrating species relative to untreated Sil1 (*Figure 2D*, lane 6 versus 5) or Sil1 lacking any thiols (*Figure 2D*, lane 9). We suggest that the minor Sil1 species migrating between the oxidized and reduced forms, observed when reduced Sil1 was treated with mal-PEG2K (*Figure 2D*, lane 6), indicates partial alkylation, wherein only one of the two Sil1 thiols is alkylated (likely a consequence of the steric hindrance for two mal-PEG2K modifications in such close proximity). Consistent with the TNB release observed spectroscopically (*Figure 2B*), treatment with mal-PEG2K indicated that BiP transitioned from an oxidized (*Figure 2D*, lane 8) to a reduced form (*Figure 2D*, lane 11) in the presence of Sil1-C203A-C373A. Importantly, in keeping with the proposed thiol-disulfide exchange reaction, Sil1 concomitantly transitioned from a reduced (*Figure 2D*, lane 8) to oxidized (*Figure 2D*, lane 11) state. Here Sil1 contains only the two N-terminal cysteines, and these data reflect redox changes in the Sil1 Cys52/Cys57 pair. The oxidation state of BiP-TNB did not change when BiP-TNB was reacted with Sil1 lacking the N-terminal cysteine pair, confirming that the N-terminal cysteines are necessary for efficient BiP-TNB reduction (*Figure 2D*, lane 9 and 12). No change in reduced Sil1 mobility was observed when Sil1 was incubated with a cysteine-less BiP (*Figure 2D*, lane 7 and 10) demonstrating that Sil1 was not becoming air-oxidized over time.

A transient high molecular weight band was also observed upon incubation of BiP-TNB and Sil1-C203A-C373A (*Figure 2D*, asterisk). The molecular weight of this band is consistent with a BiP-Sil1 mixed-disulfide intermediate (*Figure 2A*, asterisk), and interestingly the formation of the band was rapid and readily apparent immediately upon mixing (*Figure 2D*, t = 0, lane 8). These data are again consistent with a direct exchange of electrons between oxidized BiP and Sil1. Of note, a minor high molecular weight species was also seen in the BiP prep (*Figure 2D*, black circle), which we expect is a BiP-BiP disulfide-bonded dimer. We have observed a disulfide-linked BiP dimer previously in vitro with unknown significance in vivo (*Wang and Sevier, 2016*).

Intrigued by the potential visualization of the BiP-Sil1 disulfide-bonded intermediate formed during a thiol-disulfide exchange reaction (*Figure 2A*, asterisk), we sought to confirm the identity and requirements for formation of this transient species. We repeated the alkylation assay using both wild-type Sil1 and N-terminal Sil1 cysteine mutants. Here we used *N*-ethylmaleimide (NEM) as the alkylating agent, which due to its smaller size will prevent any overlapping migrating species and confounding size shifts associated with the larger mal-PEG2K. As observed with Sil1-C203A-C373A

in *Figure 2D*, a modest level of a transient high molecular weight species was observed when BiP-TNB was incubated with wild-type Sil1 (*Figure 2E*, asterisk). Use of a Sil1 single cysteine mutant (Sil1-C52A or -C57A) as reductant enhanced recovery of the mixed-disulfide species (*Figure 2E*). The stabilization ('trapping') of the BiP-Sil1 intermediate in the absence of a resolving cysteine was expected; with a single cysteine, one anticipates the attack and release of TNB (*Figure 2B*) but in the absence of a second resolving thiol, the mixed-disulfide intermediate is poorly resolved. All high molecular weight bands were resolved by reducing SDS-PAGE (*Figure 2—figure supplement 2*), confirming that these bands reflect disulfide-bonded species. Use of various BiP and Sil1 cysteine mutants established that these bands reflect the trapping of a BiP-C63–Sil1-C52/C57 intermediate (*Figure 2—figure supplements 3* and *4*). The putative BiP-Sil1 disulfide-bonded species was verified also to contain both Sil1 and BiP by immunoblotting (*Figure 2—figure supplement 5*). We attribute the late appearance of a lesser mixed-disulfide species with Sil1-C52A-C57A (that lacks significant reducing activity) to a modest (catalytically irrelevant) reactivity of the Sil1 armadillo-repeat cysteines with BiP-TNB.

Although TNB is a useful experimental proxy for physiological BiP cysteine adduct(s), BiP-TNB is not the substrate for Sil1 in cells. Thus, we sought to determine the reactivity of Sil1 towards a physiologically relevant oxidation adduct. We have shown that BiP is both sulfenylated and glutathionylated in cells (*Wang et al., 2014*; *Wang and Sevier, 2016*), and we have established conditions for BiP glutathionylation in vitro (*Wang and Sevier, 2016*). Building on our prior data, we prepared glutathionylated recombinant BiP by treating reduced BiP with a molar excess of reduced glutathione (GSH) and diamide. We followed the ability of Sil1 to reduce glutathionylated BiP by monitoring the redox state of the BiP cysteine using mal-PEG2K, which will modify BiP thiols uncovered upon glutathione removal. In order to specifically follow the reactivity of the Sil1 N-terminal cysteine pair towards glutathionylated BiP, we made use of the same Sil1 proteins as for *Figure 2D*: a Sil1-C203A-C373A mutant (with the N-terminal cysteines; CC) and a cysteine-less Sil1 (lacking the N-terminal cysteines; AA). We observed that Sil1 was able to remove the glutathione adduct from BiP, which was evident in the appearance of the slower migrating reduced BiP species over time (*Figure 3*), and that the reduction of the BiP glutathione adduct was dependent on the presence of the Sil1 N-terminal cysteines (*Figure 3*). Removal of the glutathione adduct from BiP was coincident with the oxidation of the Sil1 N-terminal cysteines, which was indicated by the appearance of a faster migrating Sil1 species over time (*Figure 3*). At present, it remains untested whether sulfenylated BiP is also a substrate for Sil1. We have shown that sulfenylated BiP can condense with GSH to yield a BiP-glutathione adduct (*Wang and Sevier, 2016*), and we have proposed that glutathionylation of BiP in cells may serve to prevent overoxidation of BiP by peroxide (the transition of a sulfenic acid adduct (–SOH) to irreversible sulfinic (–SO$_2$H) or sulfonic (–SO$_3$H) acid adducts) (*Wang and Sevier, 2016*). If Sil1 is unable to reduce a sulfenic acid adduct, we speculate that cells may require both Sil1 and glutathione to prevent irreversible oxidation of BiP.

We initially observed that a yeast strain deficient for Sil1 activity (a *sil1Δ*) showed an enhanced resistance to diamide (*Figure 1*), and we reasoned that the observed diamide resistance relates to Sil1's ability to modulate the oxidation state of BiP. Given the clear dependence for recombinant Sil1's reductant activity on its N-terminal Cys52/Cys57 pair, we expected that the diamide resistance observed for *sil1Δ* cells could be recapitulated also with cells containing a Sil1 mutant lacking the N-terminal cysteines (Sil1-C52A-C57A). In contrast, we observed that a strain possessing a *sil1-C52A-C57A* allele demonstrated the same sensitivity to diamide as a wild-type strain (*Figure 3—figure supplement 1*). These data suggest that the diamide resistance observed for a *sil1Δ* strain does not depend on Sil1 activity as a reductant, and instead that the diamide resistance observed for a *sil1Δ* strain may be a consequence of the absence of another Sil1 activity, such as a loss of NEF function. However, considering these data (*Figure 3—figure supplement 1*) alongside the diamide resistance phenotypes observed for strains in *Figure 1A and E*, we suggest a slightly altered interpretation: that the diamide resistance observed for a *sil1Δ* strain does not solely reflect a loss of Sil1 activity as a reductant or as a NEF. We propose that the diamide resistance of a *sil1Δ* strain is a byproduct of both a loss of reducant activity (and increased BiP oxidation) and also a loss of NEF activity (which may also alter BiP activity). We propose that the lack of detectable diamide resistance conferred to cells by a loss of the N-terminal Sil1 cysteines (*Figure 3—figure supplement 1*) could be a consequence of compensatory mechanism activated in the *sil1-C52A-C57A* strain.

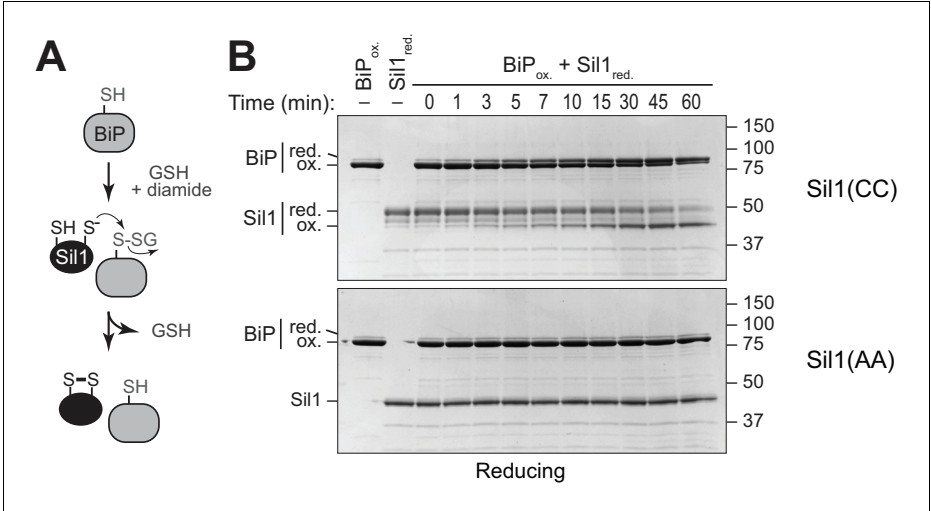

**Figure 3.** Sil1 can reduce glutathionylated BiP. (**A**) Schematic for formation and reduction of glutathionylated BiP. (**B**) Glutathionylated BiP was prepared as described in the Materials and methods. Reduction of glutathionylated BiP by Sil1-C203A-C373A (with the N-terminal cysteines; CC) or a cysteine-less Sil1 (lacking the N-terminal cysteines; AA) was monitored by following the presence or absence of free protein thiols. At the indicated times, reactions were quenched with the addition of the thiol-modifying agent mal-PEG2K, which irreversibly reacts with reduced thiols. Samples were separated by reducing SDS-PAGE, and visualized with Coomassie blue. Proteins with free thiols that become modified with mal-PEG2K show a decreased electrophoretic mobility relative to proteins with oxidized cysteines that do not react with mal-PEG2K; the oxidized and reduced forms of each protein are indicated.

The following figure supplement is available for figure 3:

**Figure supplement 1.** A yeast strain expressing a Sil1 mutant that lacks reducing activity (Sil1-C52A-C57A) does not show an increased resistance to diamide.

Struck by the potential for Sil1 to impact BiP activity both as a NEF and a reductant, we further explored the relationship between these functions. We observed that the presence of the active-site cysteines does not influence NEF activity. The inviability of a *lhs1Δ sil1Δ* strain (ascribed to a loss of NEF function) was rescued by a Sil1 catalytic-cysteine mutant (*Figure 4A*). Furthermore, both wild-type Sil1 and Sil1-C52A-C57A stimulated ATP turnover by BiP (*Figure 4B*). In contrast, no activity was observed in either assay for a Sil1-H163E mutant (*Figure 4A,B*); Sil1-H163E is defective in BiP binding (*Yan et al., 2011*) and as a consequence is ineffective as a NEF and a reductant (*Figure 2B*).

We have shown previously that recombinant oxidized BiP shows a lower steady-state ATPase rate than reduced BiP (*Wang et al., 2014*). When Sil1 activity was assessed in the context of modified BiP, we observed that only a Sil1 protein with reducing activity (wild-type Sil1) in combination with a reversible BiP-TNB substrate allowed for measurable BiP ATPase activity (*Figure 4C*). We interpret these data to reflect that wild-type Sil1 can reduce a reversible BiP modification to restore BiP ATPase activity; we expect that the restored ATPase activity of reduced BiP is, in turn, stimulated by Sil1 activity as a NEF. We observed that the activity of wild-type Sil1 or a Sil1 cysteine mutant (both functional in their ability to stimulate ATP turnover; *Figure 4B*) was insufficient to override the loss of ATPase activity observed with irreversibly oxidized BiP (*Figure 4C*, BiP-NEM). Similarly, a Sil1 protein that lacks reducing activity (Sil1-C52A-C57A) was unable to reverse the BiP-TNB adduct and restore measurable ATP hydrolysis (*Figure 4C*). In keeping with an importance for BiP-TNB reduction in restoring ATPase activity to BiP, a Sil1-H163E mutant (unable to efficiently reduce BiP-TNB; *Figure 2B*) was also unable to facilitate an increase in ATP hydrolysis for oxidized BiP (*Figure 4—figure supplement 1A*).

The ability of wild-type Sil1 to remove the BiP-TNB adduct and, in turn, stimulate ATPase activity (*Figure 4C*), is in keeping with a role for Sil1 activity in reversing and restoring BiP's chaperone function when levels of oxidative stress in the ER subside. Correspondingly, reduction of BiP-TNB by Sil1

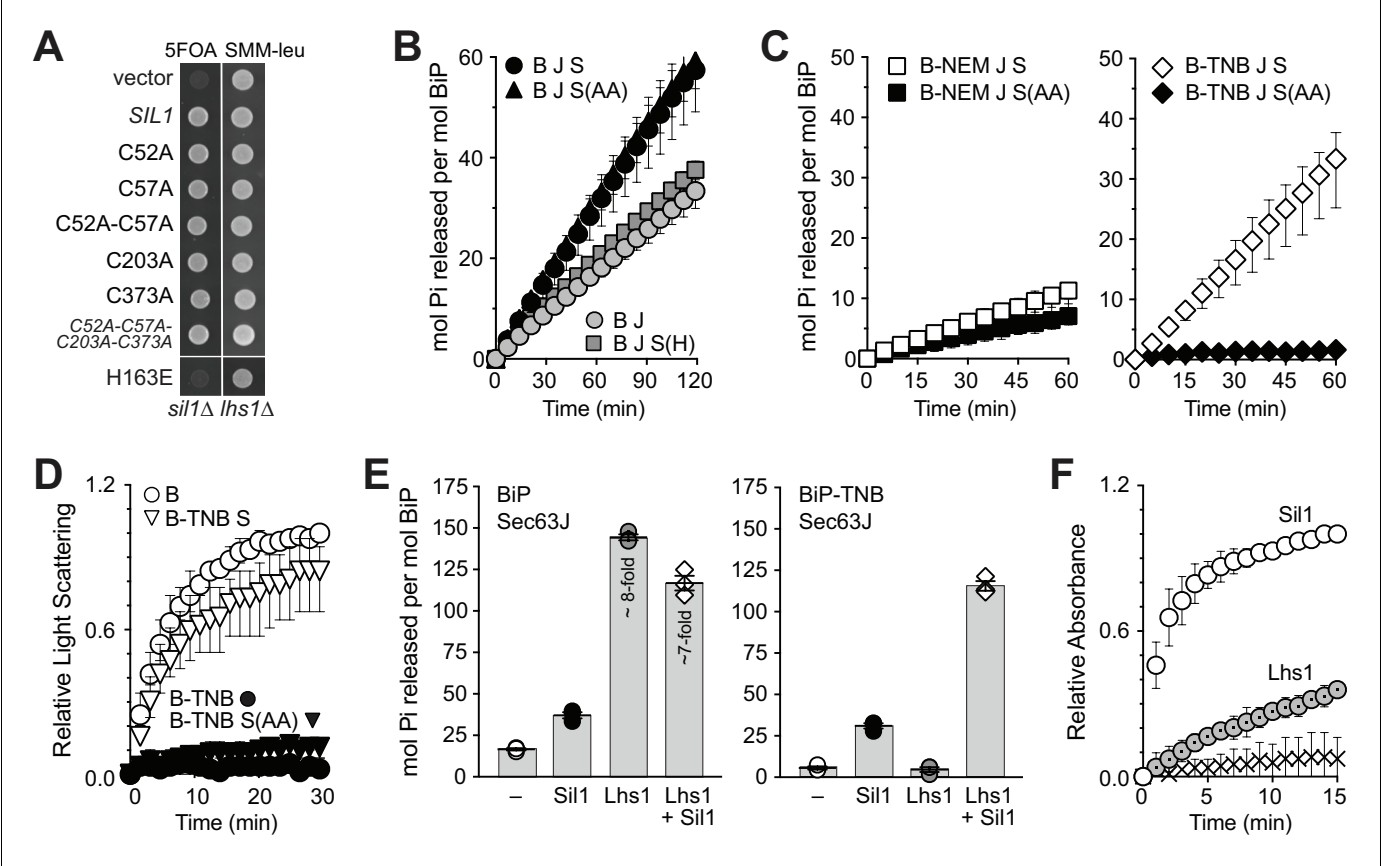

**Figure 4.** Sil1 regulates BiP activity both as a nucleotide exchange factor (NEF) and a reductant. (**A**) Sil1 cysteine mutants maintain nucleotide exchange activity. A *sil1Δ lhs1Δ* strain covered with an *URA3*-marked *SIL1* plasmid was transformed with *LEU2*-marked plasmids encoding the indicated Sil1 proteins. Their ability to substitute for wild-type Sil1 was assessed after counter-selection of the *URA3*-plasmid on 5-FOA. (**B**) BiP (B) ATPase activity was monitored by following the accumulation of free phosphate in the presence of Sec63J (J) and Sil1 (S), Sil1-C52A-C57A (S(AA)) or Sil1-H163E (S(H)). (**C**) ATP hydrolysis rates of BiP oxidized with NEM (B-NEM) or DNTB (B-TNB) was monitored in combination with J, S, or S(AA). In B and C, mean values of triplicate experiments are shown; error bars depict the range. (**D**) Aggregation of denatured rhodanese was assayed by monitoring light scattering (associated with aggregation) over time. Denatured rhodanese was diluted away from denaturant into buffer containing BiP or BiP-TNB that had been pre-incubated in the presence or absence of reduced, recombinant Sil1. Mean values of three independent experiments are shown; error bars depict the range. (**E**) The accumulation of free phosphate 15 min (left panel) or 30 min (right panel) post-ATP addition was determined for reduced BiP and BiP-TNB incubated with Sil1 (1:1 ratio) and/or Lhs1 (1:0.2 ratio) plus J-protein. Data show the mean rate of phosphate release ± SEM of three independent experiments. (**F**) Reduction of BiP-TNB by Sil1 or Lhs1 was monitored spectroscopically as in *Figure 2*. Mean values of four independent experiments are shown; error bars depict the range.

The following figure supplement is available for figure 4:

**Figure supplement 1.** Sil1-H163E only modestly reverses the decreased ATPase and increased holdase activities associated with oxidized BiP.

reversed the enhanced holdase activity also associated with modified BiP (*Figure 4D*). Consistent with our prior reports (*Wang et al., 2014*), BiP-TNB limited the aggregation of denatured rhodanese relative to unmodified BiP, evident as a decrease in light scattering observed in the presence of BiP-TNB relative to unmodified BiP (*Figure 4D*). Reduction of BiP-TNB by wild-type Sil1, but not Sil1-C52A-C57A, resulted in an aggregation profile similar to that observed with unmodified BiP (*Figure 4D*). A Sil1-H163E mutant, which shows limited reducing activity (*Figure 2B*), behaved similarly to Sil1-C52A-C57A and was unable to markedly reverse the enhanced holdase activity of BiP-TNB (*Figure 4—figure supplement 1B*).

Given the presence of both Lhs1 and Sil1 within the ER lumen, we sought to determine also how Lhs1 alone and in combination with Sil1 impacts the ATPase activity of oxidized BiP. It has been

shown previously that a BiP/J-protein/Sil1 protein mixture shows a lower steady-state ATPase rate relative to a BiP/J-protein/Lhs1 combination (*Steel et al., 2004*). We observed also that Sil1 only modestly enhanced steady-state BiP ATPase activity relative to Lhs1, which was used at an even lower concentration than Sil1 (*Figure 4E*, left panel). It has been proposed that the enhanced steady-state ATP-hydrolysis with Lhs1 reflects a reciprocal stimulation of Lhs1 ATPase activity by BiP; a difference in the ability of Sil1 and Lhs1 to release nucleotide was not observed under single-turn-over conditions (*Steel et al., 2004*). Focusing on modified BiP, we found that Lhs1 was unable to stimulate the ATPase rate of BiP-TNB, suggesting that Lhs1 is not active as a reductant. Low reducing activity was observed also when TNB release was monitored spectroscopically (*Figure 4F*). These data imply that Sil1 reducing activity is specific to Sil1 and not a common feature of the ER NEFs. Of interest, the low steady-state ATPase rate with modified BiP in the presence of Lhs1 also suggests that oxidized BiP does not appreciably stimulate Lhs1 ATPase activity. When ATPase activity of BiP-TNB was measured in the presence of both Lhs1 and Sil1, a relatively robust steady-state ATPase rate was observed (*Figure 4E*, right panel), which we attribute to the removal of the BiP-TNB adduct by Sil1 and the enhanced rate of ATP-turnover for reduced BiP that is mediated by Lhs1. These data imply a potential advantage for the presence of both Sil1 and Lhs1 in the ER following oxidative stress. Of unclear significance, we also observed a modest decrease in the overall ATPase rate when Sil1 was added to an unmodified-BiP/J/Lhs1 mixture (*Figure 4E*, left panel).

Human SIL1 shows functional and structural conservation with yeast Sil1; however, they share limited primary sequence homology (*Figure 5—figure supplement 1*). This is in contrast to mammalian and yeast BiP, which exhibit a high degree of sequence conservation (*Figure 5—figure supplement 2*). Yet strikingly, the N-terminal domain of human SIL1 contains a pair of cysteine residues, which is highly conserved between the mammalian SIL1 orthologs (*Figure 5*). The location and spacing of these cysteines in human SIL1 is similar to location and spacing of the redox-active cysteines in yeast Sil1 (*Figure 5—figure supplement 1*). Cysteine is a rare amino acid in proteins, and cysteine

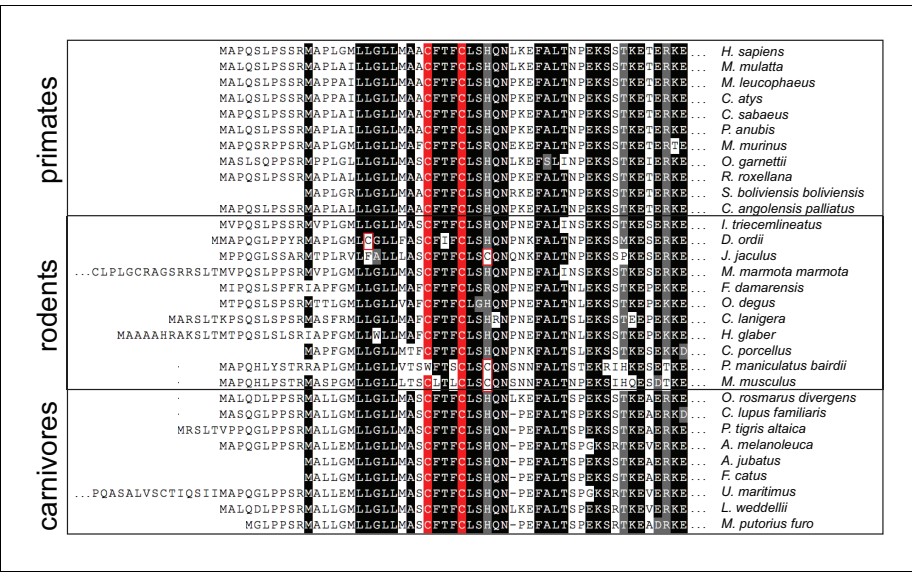

**Figure 5.** Mammalian SIL1 orthologs contain a conserved pair of cysteines within the N-terminal region. An alignment of protein sequences from SIL1 orthologs found in primates, rodents, and carnivores reveals a conserved cysteine pair in the N-terminal region. Sequences are shaded at 90% identity. Cysteines conserved between 90% of the chosen orthologs are highlighted in red; additional cysteine residues are outlined in red.

The following figure supplements are available for figure 5:

**Figure supplement 1.** Mammalian and fungal Sil1 proteins contain a conserved N-terminal cysteine pair.

**Figure supplement 2.** Sequence alignment of BiP orthologs.

conservation likely speaks to an important structural or functional role for these mammalian SIL1 cysteines (*Marino and Gladyshev, 2010*). While it remains to be determined whether human SIL1 is redox active, it is tempting to speculate that mammalian SIL1 facilitates reduction of the intramolecular disulfide described for mammalian BiP that forms in the presence of peroxide (*Wei et al., 2012*).

Mutations in human *SIL1* are associated with Marinesco-Sjögren syndrome (MSS), an autosomal recessive disorder arising in early childhood that manifests in musculoskeletal defects, cognitive delay, and early onset cataracts (*Senderek et al., 2005*; *Anttonen et al., 2005*; *Krieger et al., 2013*). Recently, a new role for SIL1 as a modifier of the neurodegenerative disorder ALS has also been described (*de Keyzer et al., 2009*). It has been assumed that MSS symptoms, and any impact of SIL1 activity in ALS progression, stem from a loss of SIL1 NEF activity and corresponding defects in protein folding and/or secretion. Yet a role for yeast Sil1 as a reductant in the ER implies a potentially important function for redox signaling in disease pathology. A majority of MSS alleles are truncations and deletions (*Goto et al., 2014*; *Ezgu et al., 2014*; *Krieger et al., 2013*), which will impact reducing activity while also destroying NEF function. It will be exciting to explore if and/or how a loss of SIL1 activity as a reductant impacts disease progression.

Currently it is unclear how Sil1 is maintained in a reduced state to facilitate BiP reduction in cells. We expect that a further understanding of the proteins and/or small molecules that donate electrons to maintain Sil1 in a reduced state (allow for Sil1 reducing activity) will provide increased insight into redox signaling within the ER. We speculate that the identification of physiological reductants for Sil1 may also point toward new candidate genetic mutations that account for the onset of disease in the 50% of MSS patients without any *SIL1* defect and no characterized genetic cause.

## Materials and methods

### Plasmid and strain construction

Plasmids are listed in *Table 1*. Yeast expression plasmids are derived from the pRS vector series (*Sikorski and Hieter, 1989*). Plasmids pJW7 and pHS116 contain *SIL1* with 720 bp of 5' and 621 bp of 3' untranslated sequence. *SIL1* and its flanking sequences were amplified from yeast genomic DNA with engineered restriction sites, and the amplified DNA was digested and ligated into compatible restriction sites in the pRS polylinker. To construct pCS637 (Sil1-His$_6$), sequence coding for a start methionine and Sil1 residues 20–407 was cloned into pET-21b, generating an in-frame fusion with sequence coding for a C-terminal His$_6$-tag. Plasmid pKP52 (GST-Sil1) was made by ligating sequence coding for Sil1 residues 22–406 into pGEX-5X-3. QuikChange mutagenesis (Agilent Technologies) was performed to generate amino acid substitution mutants using plasmid pCS757, pCS681, pJW7, pHS116, pCS817, or pCS637 as a template. All mutations were confirmed by sequencing.

Yeast strains used in this study are listed in *Table 2* and are of the S288C background. Yeast containing genomic deletions for *SIL1* and *LHS1* were obtained from the *Saccharomyces cerevisiae* genome deletion collection (*Brachmann et al., 1998*), and deletions were verified by genomic PCR. These yeast strains were backcrossed against *GAL2 ura3-52 leu2-3,112* strains from the Sevier lab collection to generate CSY448 (*sil1Δ*), CSY449 (*sil1Δ kar2-C63A*), and CSY581 (*lhs1Δ*). CSY594 and CSY595 were made by crossing CSY214 (*kar2Δ*) with CSY448 and CSY581, respectively. Strains CSY612, CSY622, CSY646, and CSY689 were generated by transformation of CSY214 with pKP37, pCS757, pCS878, or pKP97 followed by counter-selection of pCS623 on plates containing 5-fluoroorotic acid (5-FOA). Similarly, CSY625 was made by transformation of pCS757 into CSY594, and counter-selection of pCS623. Note that CSY622 is genetically equivalent to CSY318 described in *Wang and Sevier (2016)*. To generate CSY647, a heterozygous *SIL1/sil1Δ LHS1/lhs1Δ* diploid strain was transformed with pJW7, and the transformants were sporulated and tetrads were dissected. Spores containing *sil1Δ lhs1Δ* and pJW7 were unable to grow on medium containing 5-FOA.

### Yeast growth conditions

Cultures were grown in rich medium (1% Bacto-yeast extract and 2% Bacto-peptone containing 2% dextrose; YPD) or minimal medium (0.67% nitrogen base without amino acids supplemented with 16 amino acids not including cysteine) containing 2% dextrose (SMM), 2% galactose (SMM Gal) or 2%

**Table 1.** Plasmids.

| Plasmid | Description | Marker | Source |
|---------|-------------|--------|--------|
| pCS623 | KAR2 | CEN URA3 | (Wang et al., 2014) |
| pCS757 | KAR2-FLAG | CEN LEU2 | (Wang et al., 2014) |
| pCS878 | kar2-K314E-FLAG | CEN LEU2 | This study |
| pCS452 | $P_{GAL1}$-ERO1*-myc | CEN URA3 | (Sevier et al., 2007) |
| pCS681 | KAR2 | CEN LEU2 | (Wang et al., 2014) |
| pCS685 | kar2-C63A | CEN LEU2 | (Wang et al., 2014) |
| pKP37 | kar2-K314E | CEN LEU2 | This study |
| pKP97 | kar2-C63A-K314E | CEN LEU2 | This study |
| pJW7 | SIL1 | CEN URA3 | This study |
| pCS876 | sil1-C52A-C57A | CEN URA3 | This study |
| pHS116 | SIL1 | CEN LEU2 | This study |
| pKS20 | sil1-C52A | CEN LEU2 | This study |
| pKS21 | sil1-C57A | CEN LEU2 | This study |
| pKS24 | sil1-C52A-C57A | CEN LEU2 | This study |
| pKS23 | sil1-C203A | CEN LEU2 | This study |
| pKS22 | sil1-C373A | CEN LEU2 | This study |
| pCS923 | sil1-C52A-C57A-C203A-C373A | CEN LEU2 | This study |
| pCS925 | sil1-H163E | CEN LEU2 | This study |
| pCS817 | $His_6$-kar2-(42-682) | KAN | (Wang et al., 2014) |
| pCS818 | $His_6$-kar2-(42-682)-C63A | KAN | (Wang et al., 2014) |
| pKP85 | $His_6$-kar2-(42-682)-K314E | KAN | This study |
| pCS637 | sil1-(20-407)-$His_6$ | AMP | This study |
| pCS870 | sil1-(20-407)-C52A-$His_6$ | AMP | This study |
| pCS871 | sil1-(20-407)-C57A-$His_6$ | AMP | This study |
| pCS875 | sil1-(20-407)-H163E-$His_6$ | AMP | This study |
| pCS877 | sil1-(20-407)-C52A-C57A-$His_6$ | AMP | This study |
| pCS948 | sil1-(20-407)-C203A-C373A-$His_6$ | AMP | This study |
| pCS895 | sil1-(20-407)-C52A-C57A-C203A-C373A-$His_6$ | AMP | This study |
| pKP52 | GST-sil1-(22-406) | AMP | This study |
| pHS130 | $His_6$-lhs1-(21-877)-StrepII | KAN | (Xu et al., 2016) |
| pCS675 | GST-sec63J-(121-221) | AMP | (Wang et al., 2014) |

raffinose (SMM Raf). Uracil or leucine supplements were removed from minimal media to select for plasmids as needed.

## BiP mutant screen

Mutations in the BiP gene (*KAR2*) were generated by error-prone PCR using the methods described previously (*Sevier and Kaiser, 2006*) with some modifications. The entire BiP gene was amplified from pCS681 with Taq DNA Polymerase (New England Biolabs) in the presence of 0.3 mM $MnCl_2$ and an unbalanced dNTP ratio. PCR products and a gapped pRS315 vector (*Sikorski and Hieter, 1989*) were transformed into CSY278 (*kar2-C63A can1::$P_{GAL}$-ERO1\**), and yeast containing gap-repaired plasmids were isolated by selection for Leu+ transformants. ER-stress resistant transformants were identified by the ability to grow on galactose plates at 37°C.

**Table 2.** Strains.

| Strain | Genotype | Source |
|--------|----------|--------|
| CSY5 | MATa GAL2 ura3-52 leu2-3,112 | (*Wang et al., 2014*) |
| CSY214 | MATa GAL2 ura3-52 leu2-3,112 kar2Δ::KanMX [pCS623] | (*Wang et al., 2014*) |
| CSY275 | MATa GAL2 ura3-52 leu2-3,112 kar2-C63A | (*Wang et al., 2014*) |
| CSY278 | MATa GAL2 ura3-52 leu2-3,112 kar2-C63A can1::$P_{GAL1}$-ERO1*-myc | (*Wang et al., 2014*) |
| CSY289 | MATa GAL2 ura3-52 leu2-3,112 kar2Δ::KanMX [pCS681] | (*Wang et al., 2014*) |
| CSY290 | MATa GAL2 ura3-52 leu2-3,112 kar2Δ::KanMX [pCS685] | (*Wang et al., 2014*) |
| CSY612 | MATa GAL2 ura3-52 leu2-3,112 kar2Δ::KanMX [pKP37] | This study |
| CSY689 | MATa GAL2 ura3-52 leu2-3,112 kar2Δ::KanMX [pKP97] | This study |
| CSY622 | MATa GAL2 ura3-52 leu2-3,112 kar2Δ::KanMX [pCS757] | This study |
| CSY646 | MATa GAL2 ura3-52 leu2-3,112 kar2Δ::KanMX [pCS878] | This study |
| CSY448 | MATalpha GAL2 ura3 leu2 sil1Δ::KanMX | This study |
| CSY449 | MATalpha GAL2 ura3 leu2 lys2Δ0 kar2-C63A sil1Δ::KanMX | This study |
| CSY581 | MATalpha GAL2 ura3 leu2 lhs1Δ::KanMX | This study |
| CSY594 | MATa GAL2 ura3 leu2 sil1Δ::KanMX kar2Δ::KanMX [pCS623] | This study |
| CSY595 | MATa GAL2 ura3 leu2 lhs1Δ::KanMX kar2Δ::KanMX [pCS623] | This study |
| CSY625 | MATa GAL2 ura3 leu2 sil1Δ::KanMX kar2Δ::KanMX [pCS757] | This study |
| CSY647 | MATa GAL2 ura3 leu2 sil1Δ::KanMX lhs1Δ::KanMX [pJW7] | This study |

## Protein expression and purification

His$_6$-tagged BiP proteins (Kar2 residues 42–682) and GST-Sec63J protein were purified as previously described (*Wang et al., 2014*). GST-Sil1 protein (pKP52) was purified from bacteria as described for GST-Sec63J (*Wang et al., 2014*) with some adjustments. Induction of GST-Sil1 was carried out at 16°C overnight, and column washes were limited to 20 column volumes (cv) of PBS with 2 mM EDTA and 10 cv of PBS with 2 mM EDTA, 1 M KCl and 0.1% Triton-X-100. His$_6$/StrepII-tagged Lhs1 was expressed and purified as described previously (*Xu et al., 2016*), except that the initial 10 cv wash with lysis buffer was not performed, and the concentration of imidazole in the elution buffer was increased to 50 mM final.

To purify His$_6$-tagged Sil1, BL21 (DE3) cells containing the appropriate pET-derived plasmid were grown overnight at 37°C to saturation in Luria-Bertani (LB) medium with 100 μg/mL ampicillin. Cells were diluted 1:200 in LB with fresh ampicillin, and cells were grown at 37°C for 3–5 hr (until an OD$_{600}$ between 0.5 and 1.0 was reached). Cultures were shifted to 18°C, and Sil1 expression was induced with a final concentration of 0.2 mM isopropyl-$\beta$-D-thiogalactopyranoside (IPTG). Cells were harvested 16–20 hr post-induction, and cell pellets were frozen at −80°C. Pellets were solubilized in 25 mL of Sil1 lysis buffer (50 mM Na$_2$HPO$_4$ pH 7.4, 500 mM NaCl, 10 mM imidazole, 10% glycerol, 1% Triton X-100) plus one EDTA-free protease inhibitor tablet (Pierce) per 1 L of culture, and cells were lysed by treatment with lysozyme followed by sonication. Insoluble material was removed by centrifugation at 23,700 $g$ for 20 min at 4°C. Soluble material was loaded onto a HiTrap chelating column (GE Healthcare) charged with nickel. The column was washed with 100 cv of Sil1 wash buffer (50 mM Na$_2$HPO$_4$ pH 7.4, 500 mM NaCl, 20 mM imidazole, 10% glycerol), and Sil1 protein was eluted with wash buffer containing a final concentration of 0.3 M imidazole. Protein was exchanged into PBS with 10% glycerol using a PD-10 column (GE Healthcare) and concentrated to 10–30 mg/mL using a vivaspin-15 (GE Healthcare) or an Ultra-4-centrifugal filter (Amicon).

Purified proteins were flash frozen in liquid nitrogen and stored at −80°C. Concentrations were determined by BCA protein assay (Thermo Fisher Scientific) using bovine serum albumin as a standard.

## In vitro BiP activity assays

Sil1 reducing activity was measured using recombinant BiP (His$_6$-BiP) oxidized with DTNB [5,5'-dithio-bis(2-nitrobenzoic acid)] as the substrate (BiP-TNB). Recombinant His$_6$-BiP, Sil1-His$_6$, or His$_6$/StrepII-Lhs1 were each diluted to 100 µM in the same buffers used for long-term storage of these proteins at −80℃. To oxidize BiP, a 3–10-fold molar excess of DTNB was added, and samples were incubated for 1–2 hr at room temperature. To reduce Sil1 and Lhs1, proteins were incubated for 1–2 hr at room temperature in the presence of a 10-fold or greater molar excess of DTT. Unreacted DTNB and DTT were removed and buffers were exchanged using NAP-5 columns (GE Healthcare) equilibrated with TNE (10 mM Tris-HCl, pH 7.4, 50 mM NaCl, 1 mM EDTA). Catalyzed release of the TNB-adduct from oxidized BiP was measured by following the change in absorbance at 412 nm with a Beckman Coulter DU730 UV/Vis spectrophotometer. Oxidized BiP (5 µM) was incubated with 5 µM reduced Sil1 or Lhs1 in TNE, and 10 s readings were collected over 15 min. For experiments using reduced glutathione (GSH) as the reductant, oxidized BiP (5 µM) was mixed with 5–500 µM GSH. For samples containing no additional reductant (*Figure 2B,C*, X symbol, or *Figure 2—figure supplement 1*, + symbol), a DTT mixture equivalent to that used to reduce Sil1 was passed over a NAP-5 column to control for any potential DTT carryover. The initial absorbance for each reaction was set to zero. Data were normalized to a maximal value of 1.0 for wild-type Sil1 after 15 min. Graphs depict the mean normalized values from a minimum of three independent replicates. Error bars depict the range.

To follow the redox state of BiP and Sil1, an equimolar mixture of oxidized BiP and reduced Sil1 (5–20 µM each) was prepared in TNE. At the indicated time, reactions were quenched with an equal volume of buffer containing 100 mM Tris-HCl, pH 6.8, 4% SDS, 40% glycerol, 0.1% bromophenol blue and a 10-fold molar excess of *N*-ethylmaleimide (NEM), relative to the sample cysteine content. Alternatively, samples were quenched with an equal volume of 80 mM HEPES, pH 7.4, 4.8 M urea, 0.8% SDS, 20% glycerol containing a 10-fold molar excess of a 2 kDa maleimide-PEG (mal-PEG2K; Laysan Bio Inc.). Samples were incubated for 30 min at room temperature, and mal-PEG2K samples were quenched with a molar excess of free cysteine. Proteins were separated by non-reducing SDS-PAGE and visualized with a Coomassie blue stain. Samples run under reducing SDS-PAGE were supplemented with BME (5% final) prior to electrophoresis. Data shown represent a minimum of two independent assays. For detection of BiP and Sil1 by Western blotting, polyclonal antibodies raised against recombinant BiP (Kar2-(60-688)-His$_6$) or recombinant Sil1 (Sil1-(20-407)-His$_6$) were used. Antiserum to yeast BiP (RRID:AB_2636950) or Sil1 (RRID:AB_2636949) were obtained by injection of recombinant protein into rabbits by Covance Inc. (Denver, PA).

Glutathionylated BiP was prepared by reacting 50 µM BiP with 1.5 mM GSH and 750 µM diamide at 30℃ for 1 hr. Unreacted small molecules were removed using a NAP-5 column equilibrated with de-glutathionylation assay buffer (10 mM Tris-HCl, pH 8.0, 50 mM NaCl, 1 mM EDTA). For reduction assays, 5 µM glutathionylated BiP was reacted at 30℃ with 5 µM reduced Sil1 in de-glutathionylation assay buffer. At the indicated times, samples were quenched with an equal volume of 80 mM HEPES, pH 7.4, 4.8 M urea, 0.8% SDS, 20% glycerol containing a 10-fold molar excess of mal-PEG2K. After 30 min, BME was added to 5% final, and proteins were separated by reducing SDS-PAGE and visualized with a Coomassie blue stain.

ATP hydrolysis was monitored using an EnzChek Phosphate Assay Kit (Thermo Fisher Scientific) with user-supplied buffer. BiP (1 µM), GST-Sec63J (2 µM), and Sil1 (0.5 µM) were incubated in ATPase buffer (50 mM Tris-HCl, pH 7.4, 50 mM KCl, 5 mM MgCl$_2$, 1 mM DTT) with 200 µM 2-amino-6-mercapto-7-methylpurine riboside (MESG) and 0.2 U/mL purine nucleoside phosphorylase (PNP). Sample volumes were adjusted for a 96-well plate format with a final reaction volume of 200 µL. Approximately 0.3% glycerol final was also present in each reaction due to carryover from the BiP preparation. Prior to the assay, Sil1 proteins were exchanged from phosphate buffer into 2X ATPase buffer using a NAP-5 column. Reactions were initiated with the addition of 5 mM ATP final, and phosphate release was monitored at 360 nm for 1 hr with a BioTek Synergy 2 plate reader. For ATPase assays using oxidized BiP, His$_6$-BiP (100 µM) was reacted for 2 hr with a 10-fold excess of DTNB or a 50-fold excess of NEM in 10 mM Tris-HCl, pH 7.4, 50 mM NaCl, 10% glycerol. Sil1 was reduced as described above, and both BiP and Sil1 proteins were exchanged into 2X ATPase buffer without DTT using a NAP-5 column. Oxidized BiP (1.3 µM) was pre-incubated for 1 hr with GST-Sec63J (2.6 µM), Sil1 (1.3 µM), and/or Lhs1 (0.3 µM) in a final volume of 150 µL ATPase buffer lacking

DTT and containing 200 µM MESG and 0.2 U/mL PNP. Reactions were initiated with 50 µL of 20 mM ATP. Absorbance values were converted to phosphate concentrations using a phosphate standard curve. Figures show the mean phosphate turnover from a minimum of two independent experiments. Error bars depict the range.

Denatured rhodanese was prepared as described previously (*Wang et al., 2014*), except prior to the assay, rhodanese was exchanged into denaturing buffer lacking DTT using a P6 spin column (Bio-Rad). His$_6$-BiP (68 µM) was oxidized with a 30-fold excess of DTNB for 1–2 hr at room temperature, and Sil1 was reduced as described above. Each protein was exchanged into rhodanese assay buffer (20 mM HEPES-KOH, pH 7.4, 50 mM KCl) using a NAP-5 column. BiP (2 µM) and Sil1 (4 µM) were pre-incubated in rhodanese assay buffer for 1 hr at room temperature in a 96-well plate (191 µL volume), at which time 5 mM MgCl$_2$ and 1 mM ATP were added and the assay was started with the addition of 1 µM denatured rhodanese. Aggregation was monitored by following the scattering of light at 300 nm over time with a BioTek Synergy 2 plate reader. The initial timepoints were adjusted to 0 for each sample. All data were normalized to the maximal light scattering at 30 min that was observed with mock-treated BiP (*Figure 4D*) or BiP-TNB treated with Sil1 (*Figure 4—figure supplement 1B*) (set to 1.0). Data represent the mean and error bars show the range for three independent experiments.

To assess binding of recombinant BiP and Sil1, GST-Sil1 (10 µg) was incubated with 10 µl glutathione-agarose beads (Gold Biotechnology) in a total volume of 100 µL binding buffer (20 mM HEPES-KOH, pH 7.4, 100 mM KCl, 2 mM MgCl$_2$, 0.1% Igepal CA-630, 2% glycerol, 1 mM DTT, 1 µM pepstatin A). Samples were rotated for 1 hr at 4°C, and beads were collected by centrifugation at 500 *g* for 1 min. Beads were washed three times with 200 µL of binding buffer to remove unbound proteins, and washed beads were suspended in a final volume of 100 µL binding buffer. Wild-type or mutant His$_6$-BiP (10 µg) was added to the beads, and samples were rotated at 4°C for 1 hr. Beads were pelleted and washed three times with 200 µL of binding buffer. Bound proteins were solubilized in 20 µL of 2X sample buffer (100 mM Tris-HCl, pH 6.8, 4% SDS, 40% glycerol, and 0.1% bromophenol blue) containing 5% BME. Samples were resolved on a SDS-acrylamide gel and visualized with a Coomassie blue stain. Data shown represent a result observed in more than three independent assays.

## Biotin-switch assay

CSY622 and CSY625 transformed with pRS316 or pCS452 were grown to late-log phase overnight at 30°C in SMM Raf. The following morning, cells were diluted into SMM Gal, and cells were grown for 5 hr at 30°C until harvest by centrifugation. Alternatively, CSY622 and CSY646 transformed with pRS316 or pCS452 were collected after 6 hr of growth in SMM Gal at 30°C. The biotin-switch assay was performed as previously described, using BME as the reductant (*Wang et al., 2014*). The time-course assay was carried out as described in *Wang and Sevier (2016)*. In brief, CSY622 or CSY625 were grown to mid-log phase in SMM at 30°C and were treated with 5 mM diamide for 15 min. Diamide-containing medium was removed by filtration, and cells were suspended in SMM containing 20 µg/mL cycloheximide. Cells were returned to 30°C until the time of harvest. BiP-FLAG was immunoisolated from cell lysates using anti-FLAG affinity resin, which was a mixture of five parts anti-FLAG affinity resin (RRID:AB_10063035) plus one part anti-FLAG EZview anti-FLAG affinity resin (RRID:AB_2616449). Immunoblots were imaged and quantitated using a Bio-Rad ChemiDoc MP system and associated Image Lab software (RRID:SCR_014210). Biotin-labeled BiP was detected using a streptavidin-Alexa Fluor 647 conjugate (RRID:AB_2336066). BiP was visualized with a rabbit anti-BiP (Kar2) serum (RRID:AB_2636950) and a goat anti-rabbit IgG secondary antibody conjugated to an Alexa Fluor 488 (RRID:AB_2535792). Immunoblots shown are representative images, which are typical of the results obtained from a minimum of two independent experiments.

## Acknowledgements

We thank Heather Marsh for assistance with plasmid construction and protein purification. This work was supported by a National Institutes of Health grant (R01 GM105958) to CSS. Support for KDS was provided by a National Institutes of Health training grant (T32 GM007273), and KAP was supported by a National Science Foundation Graduate Research Fellowship.

## Additional information

### Funding

| Funder | Grant reference number | Author |
|---|---|---|
| National Institutes of Health | R01 GM105958 | Carolyn S Sevier |
| National Institutes of Health | T32 GM007273 | Kevin D Siegenthaler |
| National Science Foundation | Graduate Student Fellowship | Kristeen A Pareja |

The funders had no role in study design, data collection and interpretation, or the decision to submit the work for publication.

### Author contributions

KDS, Data curation, Writing—original draft; KAP, JW, Data curation, Writing—review and editing; CSS, Conceptualization, Data curation, Writing—original draft

### Author ORCIDs

Carolyn S Sevier, http://orcid.org/0000-0003-3245-6988

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
