## [Decision Letter]

Thank you for submitting your work entitled "An unexpected role for the yeast nucleotide exchange factor Sil1 as a reductant acting on the molecular chaperone BiP" for consideration by *eLife*. Your article has been favorably evaluated by a Senior Editor and three reviewers, one of whom is a member of our Board of Reviewing Editors. All three agree that the paper makes an important contribution and could be suited for publication in *eLife*, with certain revisions.

Below you will be able to read the three reviewers’ initial comments. Following this initial posting of independent critiques there was a rather lively consultation process, which yielded the following specific editorial recommendations:

Critical points (1 & 2)

1) Figure 1 shows that deletion of Sil1 imparts resistance to diamide. The mechanism proposed makes the strong prediction that re-introduction of wild type Sil1 will reverse the phenotype, imparting sensitivity, whereas a Sil1 mutant in C52A; C57A would not impart sensitivity. This experiment is rendered powerful by the fact that the authors have already shown that the C52A; C57A double mutant is functional as a NEF (as it rescues the *sil1∆;lhs1∆*; Figure 3). Furthermore, the hypothesized mechanism of action attributed to Sil1 predicts that resistance to diamide imparted by *sil1∆* should not be apparent in a Kar2-C63A genotype. If these predictions are borne out, it will greatly strengthen the conclusion that the resistance to diamide imparted by the Sil1∆ is due to loss of its redox capability and not an indirect effect of loss of NEF activity. And if the predicted outcome is not borne out, it will serve as a caveat to the reader the system is messy and future experiments challenging. Thus, we recommend that publication of a revised version be contingent on completion of these experiments and inclusion of the results, either way.

2) Given that the in vivo substrate of Sil1 is not BiP-TNB, but glutathionylated BiP, the authors should make an effort to measure Sil1's ability to reduce a BiP-glutathione adduct in vitro. The authors should attempt to confront the challenge of generating a sample of glutathionylated BiP in vitro [we considered that they might try to oxidize BiP in vitro with H2O2 in the presence of some GSH, or work with lysates of diamide-treated Sil1∆ yeast that presumably have lots of glutathionylated BiP but lack the reductase] and test the ability of exogenous reduced Sil1 to remove the glutathione or at least to render the thiol on BiP Cys63 reactive again. Here too we feel that both an outcome that supports the hypothesis and one that does not make the paper much more informative and we recommend that publication of a revised version be contingent on completion of these experiments and inclusion of the results, either way.

Non-critical points (3 & 4)

Two other experimental suggestions came up in review and we urge the authors to consider addressing them, but we advise that this should be discretional, rather than mandatory

3) The paper would be strengthened by an in vitro analysis of the impact of mutations that disrupt the Sil1/BiP interaction on the reduction reaction. However, the reviewers recognize that whilst evidence that mutations (e.g. Sil1 H163E and BiP K314E) slow down the reduction reaction in vitro would nicely support your claims, lack of a measureable effect would not constitute a sufficient challenge to render this or a similar experiment essential.

4) Documenting the formation of a mixed disulfide between Kar2p and a "resolving cysteine" mutant of Sil1p in vivo and demonstration of coherent changes in the redox status of the thiols on Sil1p in vivo would provide further strong support for the proposed mechanism. However, we recognize that the mixed disulfide and oxidized Sil1 might be ephemeral, frustrating this experiment.

We hope you find these points (and all the other reviewer's comments that follow) helpful as you prepare a revised manuscript.

Reviewer #1:

In this concise paper Sevier and colleagues make a strong case for the existence of a BiP/Kar2p-directed reductase activity of Sil1p, a protein hitherto known as a NEF for the ER-localised Hsp70 chaperone.

The study begins with an unbiased observation, whereby a mutation, K314E in BiP's NBD renders cells resistant to oxidative stress. This correlated with a higher basal level of a thiol adduct of some sort blocking the reactivity of BiP Cys63 – a residue previous discovered (by the same group) to be as an important redox regulator of BiP's activity. And by perdurance of the modification over time in cells bearing the BiP Cys63 mutation. An experiment inspired by the insight that K314 is part of the surface by which BiP engages its NEF, Sel1, showed the Sil1 deletion mimicked the BiP K314E mutation in terms the oxidation status of Cys63.

In a series of very convincing experiments conducted with purified proteins, Sevier and colleagues went on to document that Sil1 indeed can reduce a disulfide that had been introduced into BiP Cys63 in vitro. Sil1 cysteines relevant to that activity were identified and mixed disulfides between reduced Sil1 (the putative reductant) and BiP, oxidized on Cys63, were recovered in vitro. Sil1 appears to be a bit sloppy in its mechanism of action as both Sil1 Cys52 and Sil1 Cys 57 can attack the BiP disulfide, but we should hold Sevier and her colleagues to account for this "imperfection" of their enzyme.

1) Oxidation of BiP on Cys63 markedly attenuates the ability of the enzyme to chew through ATP and this is restored by Sil1. From these experiments the authors conclude that BiP oxidation scuppers nucleotide exchange. It is however unclear how the defect in π production in a multi-turnover system as that utilized in Figure 3 can distinguish between a defect in exchange and ATP hydrolysis?

2) The paper is very worthwhile as is, but would be quite glorious if the various in vitro assays in which Sil1 reduces BiP and restores it functionally to the reduced state were shown to be dependent on residues at the contact sites of Sil1 and BiP. The Sil1 H163E mutant and the BiP K314E mutant might be rather informative.

Reviewer #2:

Summary:

Sevier and colleagues identify here one of the two Bip cognate nucleotide exchange factors, SIL1, as a reductase of oxidized form of Bip (Kar2). This group previously identified a form of Kar2 that is oxidized to the sulfenic acid form at its unique Cys residue in cells overexpressing a hyperactive form of the ER oxidase Ero1 (Ero1*), the occurrence of which improves tolerance to the toxicity of Ero1* and diamide. In this study, a screen for Kar2 mutants that improve cell tolerance to Ero1* overexpression led to the identification of Kar2K314E. This mutation disrupts the interaction between Bip and Sil1, and a sil1 null mutant recapitulates the oxidative stress ER resistance of the *kar2* mutant allele, only when this allele carries its Cys residue. Based on these data they explore the potential role of Sil1 as the reductase of oxidized Kar2, in vivo by showing stabilization of this oxidized form, and in vitro using recombinant proteins. Finally, they explore the interplay between the reductase and NEF function of Sil1, and between Sil1 and the other Kar2 NEF Lhs1.

General comments:

Strength

The data presented here succeed relatively well to convince of the dual role of Sil1 as a NEF and as a reductase of Kar2, a result that is new and interesting. A few more experiments suggested below should further strengthen this conclusion.

Weakness

1) The physiological importance of Kar2 oxidation, and hence of its reduction system is not well documented. In this regards, oxidation of Kar2, and the benefit resulting from its defective reduction in terms of stress tolerance raise the question of the nature of the stress in question: authors equate the toxicity of Ero1* with that of diamide, but on which base: is ER hyperoxidation promoting non-native disulfide formation and oxidative protein misfolding? In this case, one should look at the secretion of disulfide containing substrates such as carboxipeptidase, or at least discuss the matter.

2) In a recent JBC paper, these authors showed that oxidized Kar2 become S-glutathionylated under oxidative ER stress, which raises the question of what is the actual reductant of Kar2 in vivo. In the current paper, they used DTNB in their in vitro test reductase assay: DTNB modifies Cys residue with formation of a TNB-Cys disulfide adduct resembling S-glutathionylation, which suggest that Sil1 does not reduces the sulfenic acid form of Kar2, but rather its S-glutathionylated form generated by condensation of reduced glutathione with the Cys-sulfenic acid species. Hence there might be a need of both glutathione and Sil1. This question should be addressed, at least in vitro.

Specific points:

1) By generating a sulfenic acid form of Kar2 in vitro, authors should be able to test the requirement for glutathione for Kar2 reduction by Sil1.

2) To fully demonstrate that Sil1 indeed reduces Kar2 in vivo, authors should provide an indication that the protein becomes oxidized in vivo. Perhaps, using the Cys to Ala substitution mutants, authors might be able to catch the intermolecular disulfide between Sil1 and Kar2 they see in vitro.

Reviewer #3:

This manuscript describes the finding that the nucleotide exchange factor Sil1 can reverse modification of the BiP cysteine that was shown previously to be susceptible to oxidation. In a series of logical and well-documented genetic and biochemical experiments, the authors show that Sil1 regulates BiP oxidation, with consequences for the recovery from oxidative stress conditions. Either of two cysteine residues in a presumably flexible region of Sil1 is capable of reducing BiP, but the second cysteine in the pair helps release Sil1 from mixed disulfides. The findings are important because they demonstrate how redox conditions reversibly modulate ER chaperone activity. The authors extend the implications of their findings in yeast by analyzing mammalian Sil1 sequences and highlighting the presence of an analogous pair of conserved cysteine residues.

---

## [Author Response]

Below you will be able to read the three reviewers’ initial comments. Following this initial posting of independent critiques there was a rather lively consultation process, which yielded the following specific editorial recommendations:

Critical points (1 & 2)

1) Figure 1 shows that deletion of Sil1 imparts resistance to diamide. The mechanism proposed makes the strong prediction that re-introduction of wild type Sil1 will reverse the phenotype, imparting sensitivity, whereas a Sil1 mutant in C52A; C57A would not impart sensitivity. This experiment is rendered powerful by the fact that the authors have already shown that the C52A; C57A double mutant is functional as a NEF (as it rescues the sil1∆;lhs1∆; Figure 3). Furthermore, the hypothesized mechanism of action attributed to Sil1 predicts that resistance to diamide imparted by sil1∆ should not be apparent in a Kar2-C63A genotype. If these predictions are borne out, it will greatly strengthen the conclusion that the resistance to diamide imparted by the Sil1∆ is due to loss of its redox capability and not an indirect effect of loss of NEF activity. And if the predicted outcome is not borne out, it will serve as a caveat to the reader the system is messy and future experiments challenging. Thus, we recommend that publication of a revised version be contingent on completion of these experiments and inclusion of the results, either way.

We now include data showing the diamide-resistance for a *sil1∆ kar2-C63A* strain (Figure 1) and a *sil1∆* strain expressing a Sil1-C52A-C57A mutant (Figure 3—figure supplement 1). In support of our prior prediction, the diamide resistance of a *sil1∆* strain is lessened when combined with a *kar2-C63A* mutant, suggesting a role for BiP oxidation (and Sil1 activity as a reductant) in conferring diamide resistance. However, what we observed also is that the *sil1∆ kar2-C63A* strain still shows more diamide resistance than a wild-type (or *kar2-C63A*) strain, suggesting an additional impact for loss of Sil1 activity that is independent of the impact of Sil1 on BiP oxidation. These data lead us to propose that the diamide resistance associated with a disruption in Sil1 activity in yeast is a byproduct of both a loss of reductant and NEF activities; these data are discussed in the third paragraph of the Results and Discussion. Consistent with an importance for a loss of NEF activity in conferring diamide resistance to *sil1∆* yeast, we observed that a strain containing a Sil1-C52A-C57A mutant (still active as a NEF) did not show an increase in diamide resistance relative to a strain containing wild-type Sil1; these data are shown in Figure 3—figure supplement 1 and are discussed in the thirteenth paragraph of the Results and Discussion. We suggest that the lack of any impact observed for the Sil1-C52A-C57A strain may reflect compensatory mechanisms activated in this strain. We agree with reviewers that these data are important to include and discuss, and that these results point to a "messy" system.

2) Given that the in vivo substrate of Sil1 is not BiP-TNB, but glutathionylated BiP, the authors should make an effort to measure Sil1's ability to reduce a BiP-glutathione adduct in vitro. The authors should attempt to confront the challenge of generating a sample of glutathionylated BiP in vitro [we considered that they might try to oxidize BiP in vitro with H2O2 in the presence of some GSH, or work with lysates of diamide-treated Sil1∆ yeast that presumably have lots of glutathionylated BiP but lack the reductase] and test the ability of exogenous reduced Sil1 to remove the glutathione or at least to render the thiol on BiP Cys63 reactive again. Here too we feel that both an outcome that supports the hypothesis and one that does not make the paper much more informative and we recommend that publication of a revised version be contingent on completion of these experiments and inclusion of the results, either way.

We now show the ability of Sil1 to reduce a BiP-glutathione adduct in vitro. These data are shown in a new Figure 3. We believe that these data strengthen the manuscript, confirming that Sil1 is reactive towards a physiological BiP substrate. When reporting these data, we also provide some speculation as to Sil1 activity toward the other known physiological BiP adduct, sulfenic acid (Results and Discussion, twelfth paragraph). This discussion builds upon an individual comment of reviewer 2 (point 2). We would also like to note that TNB has been used as a proxy for sulfenic acid (referenced in the fifth paragraph of the Results and Discussion), thus we do not think we can infer from the BiP-TNB data whether or not Sil1 will act on sulfenylated BiP.

Non-critical points (3 & 4)

*Two other experimental suggestions came up in review and we urge the authors to consider addressing them, but we advise that this should be discretional, rather than mandatory*

3) The paper would be strengthened by an in vitro analysis of the impact of mutations that disrupt the Sil1/BiP interaction on the reduction reaction. However, the reviewers recognize that whilst evidence that mutations (e.g. Sil1 H163E and BiP K314E) slow down the reduction reaction in vitro would nicely support your claims, lack of a measureable effect would not constitute a sufficient challenge to render this or a similar experiment essential.

We now include data that show a disruption in the reduction reaction for both a Sil1-H163E mutant (Figure 2) and a BiP-K314E mutant (Figure 2—figure supplement 1). In addition, following the individual request of reviewer 1 (point 2), we also analyze the activity of the Sil1-H163E mutant in the ATPase and rhodanese aggregation assays (Figure 4—figure supplement 1). These new data show that when the BiP-Sil1 interaction is disrupted, Sil1 is unable to act as a reductant and is unable to reverse the phenotypes associated with oxidized BiP.

4) Documenting the formation of a mixed disulfide between Kar2p and a "resolving cysteine" mutant of Sil1p in vivo and demonstration of coherent changes in the redox status of the thiols on Sil1p in vivo would provide further strong support for the proposed mechanism. However, we recognize that the mixed disulfide and oxidized Sil1 might be ephemeral, frustrating this experiment.

We whole-heartedly agree that the trapping of a BiP-Sil1 mixed-disulfide in vivo would provide further strong support for our proposed mechanism, and over the last year we have attempted (without success) to detect a Sil1-BiP mixed-disulfide species in yeast cells using a Sil1 resolving-cysteine trapping mutant. As suggested by the reviewers, we believe our inability to trap a mixed-disulfide species may reflect the transient nature of this intermediate. Similarly, we have also spent time attempting to monitor the redox state of the Sil1 thiols in vivo using a mal-PEG shift assay, like used for our in vitro studies. For these experiments, reproducible and clear controls are critical (showing a clear mobility shift between the reduced and oxidized states). To date we have had a difficult time matching mobility shifts for Sil1 isolated from yeast with the specific oxidation or reduction of the N-terminal cysteines. These are experiments we will continue to pursue.